# Multi-Head LatentMoE and Head Parallel:
# Communication-Efficient and Deterministic MoE Parallelism

**Chenwei Cui** [* 1]   **Rockwell Jackson** [* 1]   **Benjamin Joseph Herrera** [* 1]   **Ana María Tárano** [1]   **Hannah Kerner** [1]

## Abstract

Large language models have transformed many applications but remain expensive to train. Sparse Mixture of Experts (MoE) addresses this through conditional computation, with Expert Parallel (EP) as the standard distributed training method. However, EP has three limitations: communication cost grows linearly with the number of activated experts $k$, load imbalance affects latency and memory usage, and data-dependent communication requires metadata exchange. We propose Multi-Head LatentMoE and Head Parallel (HP), a new architecture and parallelism that achieve $O(1)$ communication cost regardless of $k$, completely balanced traffic, and deterministic communication, all while remaining compatible with EP. To accelerate Multi-Head LatentMoE, we propose IO-aware routing and expert computation. Compared to MoE with EP, Multi-Head LatentMoE with HP trains up to $1.82\times$ faster while having better performance. With double the granularity, the performance is even better while being $1.08\times$ faster. Our method makes multi-billion-parameter foundation model research more accessible.

 https://github.com/kerner-lab/Sparse-GPT-Pretraining

## 1. Introduction

Large language models have transformed applications such as code generation (Achiam et al., 2023), but training them remains expensive (Abnar et al., 2025). Pretraining a large model consumes substantial compute and electricity (Besiroglu et al., 2024). These costs limit who can conduct foundation model research. Reducing training costs while maintaining model quality has become a key interest.

The sparse Mixture of Experts (MoE) architecture addresses this challenge through conditional computation (Jacobs et al., 1991). For each input token, a router activates only a small subset of expert networks. This enables models to scale capacity without proportionally increasing compute (Shazeer et al., 2017; Fedus et al., 2022).

Expert Parallel (EP) is the standard distributed training method for MoEs (Lepikhin et al., 2020; Fedus et al., 2022). Each token is routed to $k$ experts, duplicated $k$ times, and sent to corresponding GPUs via all-to-all communication. Results are processed by experts and returned via another all-to-all operation (Shoeybi et al., 2019).

EP has three limitations. First, both communication volume and all-to-all latency are $O(k)$. Second, expert load imbalance further degrades all-to-all performance. Third, non-deterministic communication necessitates an additional all-to-all to exchange metadata (Shoeybi et al., 2019).

Recently, LatentMoE (Elango et al., 2026) addressed the communication bottleneck by projecting tokens from hidden dimension $d$ into a smaller latent dimension $d_h$ before communication and expert computation. This reduced both per-expert parameter loads and all-to-all traffic by a factor of $\frac{d}{d_h}$, enabling higher $k$ values at similar communication costs. However, LatentMoE still relied on Expert Parallel; communication happened after routing, so load imbalance and non-deterministic patterns persisted.

We propose Multi-Head LatentMoE and Head Parallel (HP) to address these limitations. Multi-Head LatentMoE decomposes a single MoE into multiple independent smaller modules, splitting each input token into $N_h$ sub-tokens. Each sub-token is processed by an independent MoE module with its own router and experts. HP exploits this structure. The sub-tokens are distributed to GPUs before any routing decision. Each GPU completes all routing and expert computation locally. This has three advantages over EP:

1. Communication volume is constant because each token is sent exactly once before routing.
2. Traffic is balanced because sub-tokens are evenly distributed regardless of routing decisions.
3. The inter-GPU communication is deterministic because it does not depend on routing decisions.

---

[*]Equal contribution  [1]School of Computing and Augmented Intelligence, Arizona State University, Tempe, USA. Correspondence to: Hannah Kerner <hkerner@asu.edu>.

*Proceedings of the $43^{rd}$ International Conference on Machine Learning*, Seoul, South Korea. PMLR 306, 2026. Copyright 2026 by the author(s).

A naive implementation of Multi-Head LatentMoE would multiply High Bandwidth Memory (HBM) usage and IO cost by $N_h$, because it materializes the full routing scores and expert activations for each head. We develop IO-aware routing by doing online top-$k$ directly in SRAM, reducing HBM access from $O(T \cdot N_e + N_e)$ to $O(T + N_e)$, where $T$ is the number of tokens and $N_e$ is the number of experts. We also develop IO-aware expert computation by formulating expert computation as block-sparse attention and leveraging FlexAttention (Dong et al., 2024), reducing HBM access from $O(T \cdot d_e + d_e)$ to $O(T + d_e)$, where $d_e$ is the number of neurons per expert. Both algorithms are exact.

We evaluate Multi-Head LatentMoE with HP on language modeling using 50B tokens from FineWebEdu (Penedo et al., 2024). Our method trains up to 1.82x faster than the standard MoE with EP while having 1.29 percentage points (p.p.) higher overall accuracy. When doubling the granularity, our method achieves even higher overall model performance while still being $1.08\times$ faster. Under the same number of active parameters, our method can reach up to 11.5 p.p. higher overall accuracy than MLP. The inter-GPU communication volume is reduced to 25% when $k = 4$.

Our contributions are:

1. We propose Multi-Head LatentMoE and Head Parallel, achieving $O(1)$ communication volume for any $k$, perfect load balance, and deterministic communication patterns for distributed MoE training.
2. We develop exact IO-aware routing and expert computation operations, making Multi-Head LatentMoE practical and efficient.
3. We demonstrate that Multi-Head LatentMoE with HP trains 1.82x faster than standard MoE with EP, while having better model performance. With double the granularity, it achieves even higher performance while still being $1.08\times$ faster.
4. Our work accelerates ultra-sparse MoE pre-training, making multi-billion-parameter-scale research more accessible to the academic research community.

## 2. Prerequisites

### 2.1. Notation

We use italic letters for scalars, bold lowercase for vectors, and bold uppercase for matrices or tensors. $B$ denotes batch size, $T$ denotes the number of tokens in a sequence, and $t$ indexes token position. The model hidden dimension is $d$, while $N_h$ and $d_h$ denote the number of heads and per-head dimension, respectively. For mixture-of-experts layers, $N_e$ is the number of experts, $k$ is the number of activated experts per token, and $d_e$ is the expert hidden dimension. The input and output tokens at position $t$ are $\mathbf{x}_t \in \mathbb{R}^d$ and $\mathbf{o}_t \in \mathbb{R}^d$, with projection matrices $\mathbf{W}_{\text{in}}, \mathbf{W}_{\text{out}} \in \mathbb{R}^{d \times d}$.

### 2.2. Mixture of Experts

Mixture of Experts (MoE) scales model capacity without proportionally increasing computational cost (Shazeer et al., 2017). This definition follows Mixtral (Jiang et al., 2024). We re-use this for our baseline, as well as to help define Multi-Head LatentMoE.

An MoE has $N_e$ neural network modules called experts. Among them, $k$ experts are activated for each token (Shazeer et al., 2017). For each input token $\mathbf{x}_t$, the corresponding output $\mathbf{o}_t$ is:

$$\mathbf{o}_t = \sum_{i=1}^{N_e} g_{i,t} E_i(\mathbf{x}_t), \tag{1}$$

$$g_{i,t} = \frac{\exp(g'_{i,t})}{\sum_{j=1}^{N_e} \exp(g'_{j,t})}, \tag{2}$$

$$g'_{i,t} = \begin{cases} s_{i,t}, & s_{i,t} \in \text{top-}k(\{s_{j,t} | 1 \leq j \leq N_e\}) \\ -\infty, & \text{otherwise} \end{cases}, \tag{3}$$

$$s_{i,t} = r(\mathbf{x}_t)_i. \tag{4}$$

where $g_{i,t}$ is the normalized gating value for the $i$-th expert; $E_i(\cdot)$ denotes the $i$-th expert; $s_{i,t}$ and $g'_{i,t}$ are intermediate values; top-$k(\cdot)$ selects the top $k$ values from the set; and $r(\cdot)$ is a linear mapping called the router.

To ensure router stability, our load balancing strategy is global (Qiu et al., 2025) and auxiliary-loss-free (aux-free) (Wang et al., 2024a). Following (Liu et al., 2024), we replace the first two MoE layers with Multilayer Perceptrons (MLPs) to mitigate router imbalance in early layers. The router computation is performed in FP32.

### 2.3. Expert Parallel

Expert Parallel (EP) distributes expert weights across GPUs, enabling training of large MoE models whose parameters exceed what a single GPU can hold (Lepikhin et al., 2020).

The typical execution first duplicates tokens and clusters them by target experts (Shoeybi et al., 2019). Metadata about queue lengths are exchanged via all-to-all. Afterwards, tokens are dispatched to destination GPUs through token all-to-all. Received tokens are sorted by expert index, [1] for Grouped GEMM computation. Finally, outputs are re-sorted and returned via another all-to-all, then sorted again for weighted aggregation.

EP has three limitations. First, communication volume increases with $k$ due to token duplication. Second, if expert queues have uneven lengths, all-to-all must wait for the longest queue. Third, non-deterministic communication patterns require exchanging metadata via all-to-all.

---

[1]For example, using `moe_sort_chunks_by_index(.)` from NVIDIA Transformer Engine (NVIDIA, 2022).

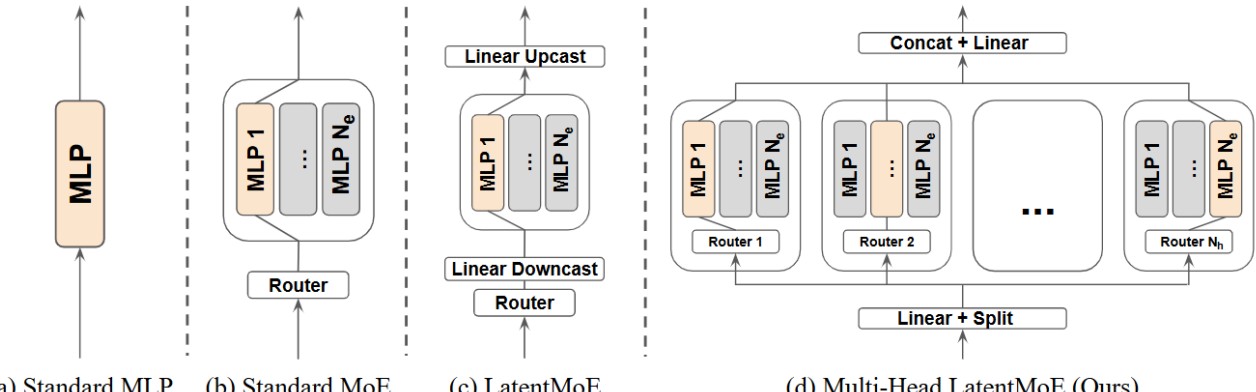

*Figure 1.* Comparison of feedforward architectures. (a) Standard MLP applies a single feedforward network to each token. (b) Standard MoE uses a router to dynamically select experts from a single set. (c) LatentMoE performs routing first, then applies linear down-projection before expert computation and linear up-projection afterward. (d) Multi-Head LatentMoE projects each token into multiple sub-tokens, each processed by an independent MoE module with its own separately-trained router and expert set (the weights are not shared). Orange blocks denote activated experts; gray blocks denote inactive experts. Black lines indicate data flow.

## 2.4. Hardware-Aware Design

Modern GPUs feature hierarchical memory with high-bandwidth low-capacity on-chip SRAM and low-bandwidth large-capacity off-chip HBM. The standard practice in architecture research is to exploit this structure.

FlashAttention (Dao, 2023) uses tile streaming and recomputation to keep intermediate results in SRAM without materializing the full attention matrix in HBM. FlexAttention (Dong et al., 2024) extends FlashAttention with block sparse attention and addresses the software lottery problem, where developers must write and maintain custom CUDA programs for novel designs.

We apply the same principle to implement IO-aware routing (see Section 3.3), keeping the intermediates in SRAM. For dropless (Gale et al., 2023) expert computation (see Section 3.4), we reuse FlexAttention to achieve highly optimized exact computation.

## 3. Method

### 3.1. Multi-Head LatentMoE

The three limitations of Expert Parallel share a common root cause: the all-to-all traffic depends on routing decisions. We need an architecture that decouples all-to-all from routing.

We propose Multi-Head LatentMoE (Figure 1). The key idea is to project a token into multiple sub-tokens, each processed by an independent MoE instance. This builds upon Latent-MoE (Elango et al., 2026) and Multi-Head MoE (Huang et al., 2024), but different from either.

Specifically, for each input token $\mathbf{x}_t$, Multi-Head Latent-MoE first projects and splits it into a list of sub-tokens:

$$[\mathbf{x}_{t,1}, \ldots, \mathbf{x}_{t,N_h}] = \text{split}(\mathbf{W}_{\text{in}}\mathbf{x}_t) \qquad (5)$$

where $\mathbf{W}_{\text{in}} \in \mathbb{R}^{d \times d}$ is a learned linear projection into the latent space, and $\text{split}(\cdot)$ divides the projected vector into $N_h$ sub-vectors of dimension $d_h$. The common practice (Vaswani et al., 2017) is to set $d_h \cdot N_h = d$, though they do not have to match.

The output of Multi-Head LatentMoE is:

$$\mathbf{o}_t = \mathbf{W}_{\text{out}} \cdot \text{concat}(f_1(\mathbf{x}_{t,1}), \ldots, f_{N_h}(\mathbf{x}_{t,N_h})) \qquad (6)$$

where $\mathbf{W}_{\text{out}} \in \mathbb{R}^{d \times d}$ projects the concatenated outputs back to the original space; each $f_i$ is an independent MoE instance as defined in Section 2.2, with its own router and expert set, sharing no parameters.

Multi-Head LatentMoE has a total FLOPs equaling that of an MoE, discounting the linear projections. However, a naive implementation would store $N_h$ sets of routing scores and expert activations, multiplying the HBM and IO cost by $N_h$. We address this challenge in Section 3.3 and Section 3.4.

### 3.2. Head Parallel

Multi-Head LatentMoE creates independent sub-tokens and MoE instances. This creates natural boundaries that divide expert weights and tokens.

We propose Head Parallel, a distributed training method for Multi-Head LatentMoE. The key idea is to move all-to-all to before routing.

Specifically, let $P$ denote the number of GPUs, where $P \leq N_h$ and $N_h$ is divisible by $P$. Initially, each GPU

---

**Algorithm 1** IO-Aware Routing: Forward Pass

---

**Require:** Sub-tokens $\mathbf{X} \in \mathbb{R}^{B \times T \times N_h \times d_h}$, router weights $\mathbf{W}_r \in \mathbb{R}^{N_h \times d_h \times N_e}$, load-balancing bias $\mathbf{b} \in \mathbb{R}^{N_h \times N_e}$, number of active experts $k$, block sizes $N$ (tokens), $M$ (experts).

**Ensure:** Top-$k$ scores $\mathbf{S}_{\text{top}} \in \mathbb{R}^{B \times T \times N_h \times k}$, top-$k$ indices $\mathbf{I}_{\text{top}} \in \mathbb{N}^{B \times T \times N_h \times k}$.

1: Divide $\mathbf{X}$ into $\lceil T/N \rceil$ blocks of $N$ tokens each.
2: **for** each $(b, t_{\text{block}}, h)$ **in parallel do**
3:     Load $\mathbf{X}_{\text{block}} \in \mathbb{R}^{N \times d_h}$ from HBM to SRAM.
4:     Initialize accumulator $\mathcal{A} \in \mathbb{N}^{N \times k}$ with 0.
5:     **for** $e_{\text{block}} = 0$ to $\lceil N_e/M \rceil - 1$ **do**
6:         Load $\mathbf{W}_{r,\text{block}} \in \mathbb{R}^{d_h \times M}$ and $\mathbf{b}_{\text{block}} \in \mathbb{R}^{1 \times M}$ from HBM to SRAM.
7:         On chip, compute $\mathbf{S}_{\text{block}} = \mathbf{X}_{\text{block}} \mathbf{W}_{r,\text{block}} + \mathbf{b}_{\text{block}} \in \mathbb{R}^{N \times M}$.
8:         On chip, pack scores and indices into 64-bit unsigned integers.
9:         On chip, find top-$k$ of $\mathbf{S}_{\text{block}}$ along expert dimension.
10:       On chip, merge with $\mathcal{A}$ and retain the best $k$.
11:     **end for**
12:     On chip, unpack $\mathcal{A}$ to scores $\mathbf{S}'_{\text{top}}$ and indices $\mathbf{I}_{\text{top}}$.
13:     On chip, compute $\mathbf{S}_{\text{top}} = \mathbf{S}'_{\text{top}} - \mathbf{b}[\mathbf{I}_{\text{top}}]$ (remove bias).
14:     Write $\mathbf{S}_{\text{top}}$ and $\mathbf{I}_{\text{top}}$ to HBM.
15: **end for**

---

**Algorithm 2** IO-Aware Routing: Backward Pass

---

**Require:** Sub-tokens $\mathbf{X} \in \mathbb{R}^{B \times T \times N_h \times d_h}$, router weights $\mathbf{W}_r \in \mathbb{R}^{N_h \times d_h \times N_e}$, top-$k$ indices $\mathbf{I}_{\text{top}} \in \mathbb{N}^{B \times T \times N_h \times k}$, gradient of top-$k$ scores $d\mathbf{S}_{\text{top}} \in \mathbb{R}^{B \times T \times N_h \times k}$, block size $N$ (tokens).

**Ensure:** Input gradient $d\mathbf{X} \in \mathbb{R}^{B \times T \times N_h \times d_h}$, router weight gradient $d\mathbf{W}_r \in \mathbb{R}^{N_h \times d_h \times N_e}$.

1: Initialize $d\mathbf{W}_r = \mathbf{0}$ in HBM.
2: Divide $\mathbf{X}, \mathbf{I}_{\text{top}}, d\mathbf{S}_{\text{top}}$ into $\lceil T/N \rceil$ blocks of $N$ tokens each.
3: **for** each $(b, t_{\text{block}}, h)$ **in parallel do**
4:     Load $\mathbf{X}_{\text{block}} \in \mathbb{R}^{N \times d_h}$ from HBM to SRAM.
5:     Initialize $d\mathbf{X}_{\text{accum}} = \mathbf{0} \in \mathbb{R}^{N \times d_h}$ on chip.
6:     **for** $i = 1$ to $k$ **do**
7:         Load $\mathbf{e}_i = \mathbf{I}_{\text{top}}[:,:,:,i] \in \mathbb{N}^N$ and $d\mathbf{s}_i = d\mathbf{S}_{\text{top}}[:,:,:,i] \in \mathbb{R}^N$ from HBM to SRAM.
8:         Load $\mathbf{W}_r[:,\mathbf{e}_i] \in \mathbb{R}^{N \times d_h}$ from HBM to SRAM (gather by expert index).
9:         On chip, compute $d\mathbf{X}_{\text{accum}} \mathrel{+}= d\mathbf{s}_i \cdot \mathbf{W}_r[:,\mathbf{e}_i]$.
10:       Atomic add $\mathbf{X}_{\text{block}}^\top d\mathbf{s}_i$ to $d\mathbf{W}_r[:,\mathbf{e}_i]$ in HBM.
11:     **end for**
12:     Write $d\mathbf{X}_{\text{accum}}$ to $d\mathbf{X}$ in HBM.
13: **end for**

---

holds a tensor of sub-tokens with shape $(B, T, N_h, d_h)$. All-to-all redistributes this tensor so that each GPU receives all sub-tokens for its assigned heads, yielding shape $(B, T, P, N_h/P, d_h)$. After MoE computation, a reverse all-to-all sends the outputs back to the original GPUs.

Head Parallel is *not* fundamentally bounded by $N_h$. Heads can be replicated across multiple GPUs when scaling beyond $N_h$, similar to how EP can replicate experts across nodes to improve parallelizability. For instance, DeepSeek-V3 (Liu et al., 2024) trains 256 experts on 2048 GPUs. Also, HP composes naturally with other parallelism strategies, such as Expert Parallel, to scale beyond $N_h$ GPUs. The key insight is that HP provides a communication-efficient highway for distributing sub-tokens, which can serve as a building block within larger distributed training systems.

Head Parallel streamlines token communication. First, each token is sent exactly once without duplication, so communication volume is $O(1)$ relative to $k$. Second, each GPU sends and receives the same amount of data deterministically, so there is no need to communicate any metadata beforehand. This also avoids edge cases such as waiting for the slowest node or out-of-memory, when too many tokens route to a single GPU. We verify these advantages in Section 4.2.

### 3.3. IO-Aware Routing

Multi-Head LatentMoE projects each token into $N_h$ sub-tokens. However, a naive router materializes the routing scores in HBM before selecting the top-$k$ experts. With $N_h = 8$, the HBM and IO costs are 8 times higher.

We propose IO-aware routing, inspired by FlashAttention (Dao et al., 2022). The key observation is that a router only outputs the top-$k$ indices and scores. Therefore, we can stream the top-$k$ results in SRAM without materializing in HBM.

The forward pass is given in Algorithm 1. For each block of experts, we compute scores in SRAM and find the block's local top-$k$ results. The local results are merged with an accumulator to track the running top-$k$ indices and scores, which are written to HBM by the end.

It is worth mentioning two computational tricks. First, to support aux-free load balancing (Wang et al., 2024a), we add bias before top-$k$ and subtract it afterward. As such, bias influences routing without affecting the final scores. Second, following Tillet et al. (2019), we pack the score and index into a 64-bit unsigned integer to achieve arg-top-$k$ in Triton.

The backward pass is given in Algorithm 2. Given $\mathbf{S} = \mathbf{XW}$, the input gradient is $d\mathbf{X} = d\mathbf{S} \cdot \mathbf{W}^\top$ and the weight gradient is $d\mathbf{W} = \mathbf{X}^\top \cdot d\mathbf{S}$. Gradients are computed only for the $k$ selected experts per token, reducing time complexity

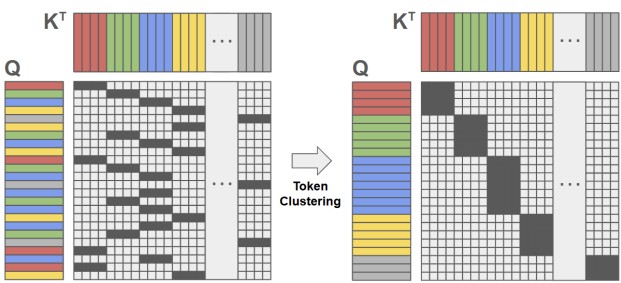

*Figure 2.* Token clustering for expert computation expressed as block-sparse attention. Q represents input tokens and K transpose represents expert weights. Colors encode expert assignments. Clustering reorders Q so that tokens assigned to the same expert become contiguous, yielding a block-diagonal sparsity pattern.

from $O(N_e)$ to $O(k)$, given that $k << N_e$.

IO-aware routing is efficient and exact. For both algorithms, the HBM access reduces from $O(N_e)$ to $O(k)$. The forward pass has the same number of FLOPs as a standard router. The backward pass exploits sparsity, reducing its time complexity from $O(N_e)$ to $O(k)$. This makes our router more efficient than a standard router. We verify these claims in Section 4.2.

### 3.4. IO-Aware Expert Computation

Multi-Head LatentMoE projects each token into $N_h$ sub-tokens. However, Grouped GEMM (NVIDIA, 2022) materializes the expert activations, making memory grow linearly with $N_h$. We need an operator that is dropless (Gale et al., 2023) like Grouped GEMM, but also IO aware.

We propose IO-aware expert computation. Observing the well-known duality between FFN and Attention (Vaswani et al., 2017), we rewrite sparse expert computation into block sparse attention. This allows us to leverage FlexAttention (Dong et al., 2024), reusing their highly-optimized IO-aware kernels.

FlexAttention (Dong et al., 2024) extends FlashAttention with a score modification function and a block sparse mask:

$$\mathbf{O} = \text{softmax}(\text{score\_mod}(\mathbf{QK}^\top \odot \mathbf{M})) \cdot \mathbf{V} \quad (7)$$

where $\text{score\_mod}(\cdot)$ is a user-defined function applied element-wise, and $\mathbf{M}$ is a binary block sparse mask. FlexAttention also returns $\log(\ell)$, where $\ell$ is the softmax denominators. We leverage $\ell$ to change the activation function.

Expert computation can be written as block sparse attention. Specifically, a single expert computes $\sigma(\mathbf{XW}_{\text{in}}^\top)\mathbf{W}_{\text{out}}$, which mirrors the blockwise $\text{softmax}(\mathbf{QK}^\top)\mathbf{V}$.

To switch from softmax to gelu, consider

$$\text{score\_mod}(s) = \log(\text{gelu}(s) + 1),$$

where offsetting by 1 makes the logarithm well-defined. [2] The softmax output then becomes

$$\frac{\exp(\log(\text{gelu}(s) + 1))}{\sum_j \exp(\log(\text{gelu}(s_j) + 1))} = \frac{\text{gelu}(s) + 1}{\ell}. \quad (8)$$

We note that $\ell$ is numerically stable.

FlexAttention returns both $\log(\ell)$ and

$$\mathbf{O}' = (\text{gelu}(\mathbf{XK}^\top) + 1)\mathbf{V}/\ell.$$

Since $\ell$ is numerically stable, we can materialize it. Multiplying $\mathbf{O}'$ by $\ell$ yields

$$(\text{gelu}(\mathbf{XK}^\top) + 1)\mathbf{V} = \text{gelu}(\mathbf{XK}^\top)\mathbf{V} + \mathbf{1V}.$$

Finally, subtracting the bias term $\mathbf{1V} = \sum_j \mathbf{V}_j$ for each token's assigned expert recovers the exact result. This can be implemented with `torch.gather`.

Similar to Grouped GEMM, our expert computation requires tokens assigned to the same expert placed contiguously.

This is usually through token clustering (See Figure 2).

The implementation is pure-PyTorch (Paszke et al., 2019). Practitioners with a limited engineering budget can avoid writing custom CUDA kernels or expanding support for new hardwares.

IO-aware expert computation is dropless (Gale et al., 2023) and numerically correct like Grouped GEMM, but also IO-aware. We verify these claims in Section 4.2.

## 4. Experiments

### 4.1. Language Modeling

Multi-Head LatentMoE aims to match or exceed traditional MoE in model quality while providing faster training. In this section, we evaluate both model performance and training efficiency under controlled computation budgets.

We conduct language modeling experiments on a 50B token subset of FineWeb-EDU (Penedo et al., 2024). For detailed experimental setup including model hyperparameters, see Appendix A and Appendix B. For 10B token experiments, see Appendix C. For comparison with DeepEP (Zhao et al., 2025b), see Appendix D.

Table 1 shows the results. Multi-Head LatentMoE HP achieves comparable model performance to baselines while training significantly faster: 1.12x faster at 2B total parameters (157.94 vs 177.09 hours) and 1.82x faster at 4B total parameters (174.07 vs 316.34 hours). This speedup comes

---

[2]For activation functions that are unbounded below, one can compute the offset online, but this is beyond our scope.

*Table 1.* Model performance comparison across feedforward designs. Parameters are reported as activated-total. FineWebEDU reports validation perplexity (ppl.). HellaSwag (Hella.), PiQA, LAMBADA (LMB.), ARC-Easy (Arc E.), and ARC-Challenge (Arc C.) report zero-shot accuracy (acc.). G doubles the granularity. Bold indicates best and underline indicates second best. LatentMoE is replicated based on Elango et al. (2026). All accuracies are normalized based on length, except for LAMBADA.

| Parameters active - all | Model | Training Cost hours (rel.) ↓ | FineWebEDU ppl. ↓ | Hella. acc. ↑ | PiQA acc. ↑ | LMB. acc. ↑ | Arc E. acc. ↑ | Arc C. acc. ↑ | Avg. acc. ↑ |
|---|---|---|---|---|---|---|---|---|---|
| 0.2B-0.2B | MLP | 55.03 (1.00×) | 17.63 | 35.3 | 63.2 | 30.1 | 49.6 | 26.7 | 40.99 |
| 0.2B-2.2B | MoE EP | 177.09 (1.00×) | 13.12 | 46.7 | 70.2 | 37.4 | 59.8 | 29.4 | 48.70 |
| | LatentMoE EP | 158.88 (0.90×) | 12.80 | 48.2 | 70.7 | 37.1 | 56.1 | 31.4 | 48.72 |
| | **MH LatentMoE HP** | **157.94 (0.89×)** | 12.84 | 48.2 | 70.8 | 38.0 | 59.5 | 32.8 | 49.86 |
| | **MH LatentMoE HP G** | 219.79 (1.24×) | **12.59** | 49.5 | 70.9 | 39.0 | 60.6 | 32.4 | **50.49** |
| 0.2B-4.2B | MoE EP | 316.34 (1.00×) | 12.54 | 49.2 | 70.6 | 38.0 | 60.6 | 31.7 | 50.00 |
| | LatentMoE EP | 270.57 (0.86×) | 12.31 | 50.1 | 71.3 | 38.1 | 61.7 | 32.1 | 50.66 |
| | **MH LatentMoE HP** | **174.07 (0.55×)** | 12.28 | 50.1 | 71.4 | 40.2 | 60.5 | 34.2 | 51.29 |
| | **MH LatentMoE HP G** | 293.45 (0.93×) | **11.99** | 51.1 | 72.3 | 39.8 | 63.6 | 35.7 | **52.49** |

from two sources. First, Head Parallel achieves O(1) communication volume compared to $O(k)$ for Expert Parallel; with k=4, our communication volume is 25% of Expert Parallel. Second, our FlexAttention-based expert computation is faster under high sparsity, likely due to different work partitioning and FlexAttention's well-optimized kernels.

We also replicate the finding from Elango et al. (2026) that LatentMoE EP trains faster than MoE EP. However, our method is faster while achieving better or comparable performance. We attribute this to the fact that our method is fundamentally EP-free, requiring only HP.

Following Elango et al. (2026), we reinvest the speedup advantage into doubled granularity (MH LatentMoE HP G). At the 4B scale, MH LatentMoE HP G achieves the best average accuracy (52.49%) while still training 1.08x faster than MoE EP (293.45 vs 316.34 hours).

These results demonstrate that Multi-Head LatentMoE provides significant training speedups over MoE. When reinvesting the saved compute into increased granularity, Multi-Head LatentMoE achieves both faster training and better model quality.

### 4.2. Per-Component Analysis

The training speedup of Multi-Head LatentMoE with Head Parallel comes from three factors: reduced all-to-all communication, efficient routing, and efficient expert computation. In this section, we isolate each component to verify its contribution.

We first compare Expert Parallel (EP) and Head Parallel (HP) under varying load imbalance. We measure all-to-all latency (including time spent waiting for the slowest GPU) and peak VRAM usage on 4 GPUs. We control load imbalance by drawing per-token expert assignments from a Zipf distribution with skew in $\{0.0, 1.0, 2.0\}$, where

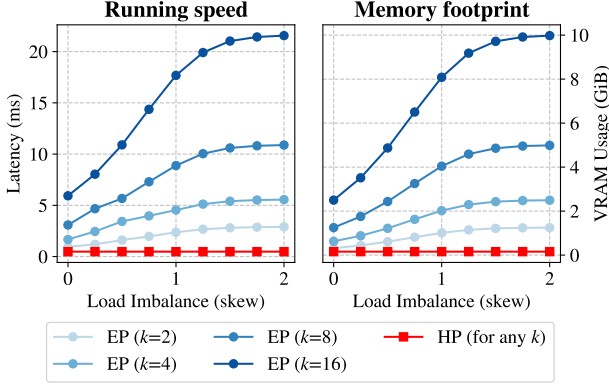

*Figure 3.* Comparison of Expert Parallel (EP) and Head Parallel (HP) under varying load imbalance. Left: all-to-all latency, including time spent waiting for the slowest GPU. Right: peak VRAM usage across all GPUs. We simulate load imbalance on 4 GPUs using a Zipf distribution with varying skew: skew=0.0 corresponds to uniform distribution (25% per GPU), skew=1.0 assigns 80.8% of tokens to GPU 0, and skew=2.0 assigns 99.8% to GPU 0. EP latency and memory grow with both $k$ and skew, while HP remains constant for any $k$.

skew=0.0 corresponds to uniform distribution and skew=2.0 assigns 99.8% of tokens to a single GPU. Figure 3 shows that EP latency and peak VRAM both increase with larger $k$ and larger skew. This is expected because EP duplicates and redistributes tokens after routing, and must wait for the most-loaded device. In contrast, HP remains constant across all values of $k$ and skew because each token is sent exactly once before routing, making traffic per GPU fixed.

We next compare naive routing (using `torch.matmul`) with our IO-aware routing. We measure forward and backward latency and combined memory footprint, using $B = 40$, $T = 2048$, $N_h = 8$, $d_h = 128$, and varying the number of experts. Figure 4 shows that our IO-aware routing

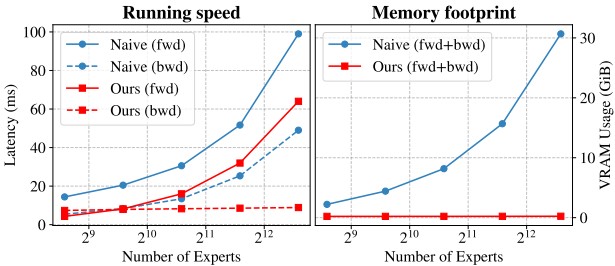

*Figure 4.* Comparison of naive routing with `torch.matmul` and our IO-aware routing. Left: latency for forward and backward passes. Right: memory footprint for forward and backward combined. Our IO-aware routing maintains constant memory footprint regardless of the number of experts, and its backward pass remains nearly constant due to sparse gradient computation. Experiments use $B = 40$, $T = 2048$, $N_h = 8$, $d_h = 128$.

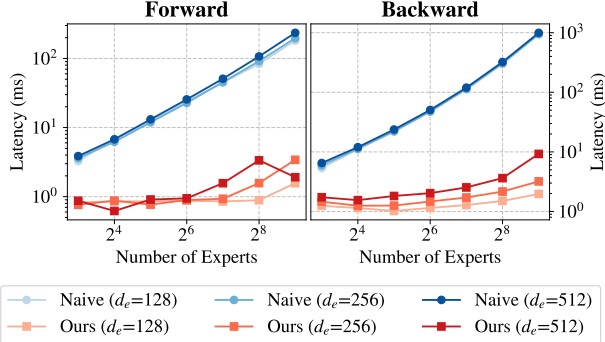

*Figure 5.* Comparison of naive expert computation (using grouped GEMM) and our IO-aware expert computation under Multi-Head LatentMoE, where multiple heads increase the number of sub-tokens. Left: forward pass latency. Right: backward pass latency (log scale). $d_e$ denotes the number of hidden neurons per expert. Naive grouped GEMM scales poorly with the number of experts, while our method remains efficient across all configurations. Experiments use $B = 4$, $T = 512$, $N_h = 8$, $d_h = 128$, $k = 4$.

maintains constant memory footprint as the number of experts grows, while the naive implementation scales linearly. The backward pass of our method remains nearly constant because gradients are computed only for the selected top-$k$ experts.

Finally, we compare naive expert computation (using grouped GEMM) with our FlexAttention-based approach under Multi-Head LatentMoE, where multiple heads increase the number of sub-tokens. We measure forward and backward latency using $B = 4$, $T = 512$, $N_h = 8$, $d_h = 128$, $k = 4$, and varying expert size $d_e$ and expert count. Figure 5 shows that naive grouped GEMM scales poorly with the number of experts, while our method remains efficient across all configurations. The gains are particularly large in the backward pass.

These results validate that Head Parallel removes the sensitivity to load imbalance inherent in Expert Parallel, and that our IO-aware routing and expert computation make Multi-Head LatentMoE practical by avoiding excessive HBM usage while remaining performant.

### 4.3. Training-Time Profiling

To isolate the source of end-to-end speedup, we profile forward and backward latency for each stage. We compare MoE with Expert Parallel against Multi-Head LatentMoE with Head Parallel, measured on 4 H100 GPUs with 2B parameters. Table 2 reports stage-wise latency, and Table 3 attributes the overall speedup to each component.

*Table 2.* Stage-wise latency in milliseconds comparing MoE with Expert Parallel against Multi-Head LatentMoE with Head Parallel (Ours). fwd means forward pass, bwd means backward pass, "both" means forward and backward combined. A2A means the all-to-all operation. Expert Comp. means expert computation. Token A2A Inv. is the inverse operation of Token A2A. MH Routing is our IO-aware multi-head routing.

| System | Stage | fwd | bwd | both |
|---|---|---|---|---|
| MoE EP | Routing | 1.04 | 0.68 | 1.72 |
| | Metadata A2A | 0.12 | 0.05 | 0.17 |
| | Token A2A | 0.98 | 0.92 | 1.90 |
| | Expert Comp. | 13.06 | 22.40 | 35.45 |
| | Token A2A Inv. | 0.92 | 0.93 | 1.85 |
| Ours | Token A2A | 0.43 | 0.40 | 0.84 |
| | MH Routing | 2.35 | 4.17 | 6.52 |
| | Expert Comp. | 4.55 | 13.85 | 18.40 |
| | Token A2A Inv. | 0.49 | 0.46 | 0.95 |

*Table 3.* Speedup attribution by component. Speedup is the ratio of MoE EP latency to Ours (Multi-Head LatentMoE with Head Parallel). Difference means Ours minus MoE EP in milliseconds (negative values indicate Ours is faster). Communication involves Metadata A2A, Token A2A, and Token A2A Inv. for MoE EP, and Token A2A and Token A2A Inv. for Ours.

| Component | Speedup | Difference |
|---|---|---|
| Communication | 2.20× | −2.13 |
| Routing | 0.26× | +4.80 |
| Expert Comp. | 1.93× | −17.05 |

The profiling reveals three findings. First, Head Parallel achieves 2.20× communication speedup by eliminating metadata all-to-all and reducing token volume by $k$ times. Second, even though multi-head routing incurs 4.80 ms overhead, it is still necessary as shown in Figure 4. Third, expert computation achieves 1.93× speedup because FlexAttention's work partitioning better suits the tall and skinny matrices in ultra-sparse settings compared to GroupedGEMM.

Overall, the 17.05 ms saved in expert computation and 2.13

ms saved in communication outweigh the 4.80 ms routing overhead, yielding significant net end-to-end speedup. This confirms that the majority of observed speedup is attributable to expert computation, with communication providing additional gains. The contribution of Head Parallel is expected to be more visible in multi-node settings where communication dominates (Soboleva & Anthony, 2025; Jin et al., 2026).

### 4.4. Ablation: Head Configuration

In Multi-Head LatentMoE, each token is projected into $N_h$ sub-tokens of $d_h$ dimensions. We investigate how $N_h$ and $d_h$ affect model performance and training efficiency. The models are trained on 2.5B tokens.

We measure validation loss and training cost. Following common practice (Vaswani et al., 2017), we keep $N_h \cdot d_h = d$ and evaluate $(N_h, d_h) \in \{(16, 64), (8, 128), (4, 256)\}$.

Table 4 shows the results. The $8 \times 128$ configuration achieves the lowest training cost while maintaining competitive validation loss. The $4 \times 256$ configuration achieves the lowest validation loss of 3.41, but incurs higher training cost and high SRAM pressure. The $16 \times 64$ configuration has the lowest SRAM pressure but the highest training cost and validation loss.

The experiments show Multi-Head LatentMoE is robust to head configuration choices. Therefore, we select the $8 \times 128$ configuration as it provides the best training efficiency and offers a good balance for model performance and ease of engineering.

*Table 4.* How different head configurations affect Multi-Head LatentMoE. $N_h$ denotes the number of heads and $d_h$ denotes the head dimension. Training Cost is relative to the $8 \times 128$ configuration. SRAM Pressure refers to the on-chip memory pressure during routing and expert computation under typical work partitioning. Bold indicates best while underline indicates second best.

| $N_h \times d_h$ | Val Loss ($\downarrow$) | Training Cost (Relative $\downarrow$) | SRAM Pressure ($\downarrow$) |
|---|---|---|---|
| $16 \times 64$ | 3.56 | $1.34\times$ | **Low** |
| $8 \times 128$ | 3.48 | **1.00**$\times$ | Medium |
| $4 \times 256$ | **3.41** | 1.07$\times$ | High |

### 4.5. Ablation: Separate Routing Tokens

In Multi-Head LatentMoE, the input token $\mathbf{x}_t$ is projected into sub-tokens $\mathbf{x}_{t,i}$ that are used for both routing and expert computation. In this section, we assess an alternative design choice: projecting a separate set of sub-tokens $\mathbf{r}_{t,i}$ dedicated to routing decisions. The models are trained on 2.5B tokens.

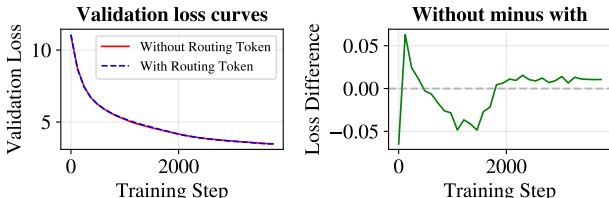

*Figure 6.* Comparing Multi-Head LatentMoE with and without separate sub-tokens for routing decisions. Left: validation loss curves over training steps. Right: loss difference (without minus with), where negative values favor the variant with separate routing tokens.

Specifically, a single token $\mathbf{x}_t$ becomes

$$[\mathbf{x}_{t,1}, \ldots, \mathbf{x}_{t,N_h}, \mathbf{r}_{t,1}, \ldots, \mathbf{r}_{t,N_h}] = \mathrm{split}(\mathbf{W}_{\mathrm{in}}\mathbf{x}_t) \quad (9)$$

where $\mathbf{W}_{\mathrm{in}} \in \mathbb{R}^{2d \times d}$ is a learned linear projection, and split($\cdot$) divides the projected token into $2N_h$ sub-tokens of dimension $d_h$. Each head $i$ then uses $\mathbf{r}_{t,i}$ for computing routing scores and $\mathbf{x}_{t,i}$ for expert computation. Note that this design doubles the all-to-all volume under Head Parallel, as both $\mathbf{x}_{t,i}$ and $\mathbf{r}_{t,i}$ must be distributed across devices.

To investigate whether this design choice improves model quality, we pretrain Multi-Head LatentMoE with and without separate routing tokens and measure validation perplexity throughout training.

Figure 6 shows the results. Separate routing tokens provide a small improvement in early training that diminishes as training progresses. Both configurations achieve nearly identical perplexity. Given that this design also doubles the all-to-all volume, we do not use separate routing tokens.

## 5. Related Work

**Latent Representations in MoE** Projecting a token into a lower-dimensional latent space is a common design choice in MoEs for both routing and expert computation.

X-MoE (Chi et al., 2022) addressed representation collapse by routing in a low-dimensional space, but experts still operated on high-dimensional inputs. Multi-Head MoE (MH-MoE) (Huang et al., 2024) linearly projected a single token into multiple smaller latent spaces, creating sub-tokens. Different from our work, these sub-tokens shared the same router and set of experts, so MH-MoE could not benefit from Head Parallel. MH-MoE did not propose IO-aware routing or expert computation. All activations are materialized in HBM. Recently, LatentMoE (Elango et al., 2026) reduced all-to-all communication volume by performing expert computation in a low-dimensional latent space, but still relied on EP and inherited its issues with load imbalance and non-deterministic communication.

Different from both Multi-Head MoE (MH-MoE) (Huang et al., 2024) and LatentMoE (Elango et al., 2026), Multi-Head LatentMoE assigns each head an independent router and set of experts, forming fully independent MoE modules. Only this design can benefit from Head Parallel.

**Distributed Training**  Distributed training in MoE partitions expert weights across devices to scale model capacity. Its communication overhead is the primary bottleneck in MoE training (Fedus et al., 2022).

GShard (Lepikhin et al., 2020) established Expert Parallel (EP), but its all-to-all operation had $O(k)$ communication volume, was affected by load imbalance, and was non-deterministic. DeepEP (Zhao et al., 2025b) improved EP through kernel fusion, communication-computation overlap, and avoided sending duplicate tokens to the same node. However, without fundamentally changing the parallelism, DeepEP only reduced constant factors without changing the $O(k)$ complexity. Tensor Parallel (Shoeybi et al., 2019) provided balanced and deterministic communication, but had large communication volume. In attention research, Ulysses (Jacobs et al., 2023), also known as Head Parallel, distributed attention heads across devices to enable long sequence modeling. Our work is fundamentally different.

Our Head Parallel distributes expert weights across GPUs like EP, but moves all-to-all before routing. It is the first MoE parallelism achieving $O(1)$ communication volume along with balanced and deterministic communication.

**Efficient MoE Systems**  Recent work has explored various approaches to improve MoE efficiency at scale.

DeepSeek-V3 (Liu et al., 2024) employs finer-grained experts with auxiliary-loss-free load balancing and shared experts, but still relies on Expert Parallel for distributed training. AdapMoE (Zhong et al., 2024) reduces inference overhead through adaptive expert gating that adjusts the number of activated experts based on sensitivity, targeting edge deployment rather than distributed training. Deep-GEMM (Zhao et al., 2025a) provides optimized CUDA kernels for MoE computation including fused MoE with overlapped communication, but its optimizations are at the kernel level rather than changing the parallelism strategy. Occult (Luo et al., 2025) optimizes communication across experts after routing through collaborative scheduling; our Head Parallel optimizes communication before routing, making them orthogonal and potentially combinable.

Our work differs from these system-level optimizations by changing the fundamental parallelism strategy itself, achieving $O(1)$ communication complexity rather than reducing constant factors.

## 6. Conclusion

In this work, we addressed the fundamental limitations of EP with Multi-Head LatentMoE and HP. To make Multi-Head LatentMoE practical, we developed new routing operators using online top-$k$ and FlexAttention-based expert computation, achieving FlashAttention-like IO awareness while being exact.

Head Parallel achieves $O(1)$ communication volume regardless of the number of activated experts, perfectly balanced traffic across GPUs, and deterministic communication patterns. Being static means HP does not require communicating metadata between GPUs. Experiments show that Multi-Head LatentMoE with HP trains up to 1.82x faster than traditional MoE. The communication volume reduced to 25% of EP at $k = 4$. With doubled granularity, it achieves higher overall performance while still being 1.08x faster.

Our work accelerates ultra-sparse MoE training, making multi-billion-scale foundational research more accessible.

**Scope, limitations, and future directions.** Our approach is most effective in the ultra-sparse regimes with large expert counts and small expert sizes, which characterize modern MoE architectures (Yang et al., 2025). Scalability at production-relevant scale remains the central open question. In our current single-node, compute-bound setting, the majority of observed speedup is attributable to IO-aware expert computation rather than reduced communication from Head Parallel alone. The contribution of Head Parallel is expected to be more visible in multi-node settings where communication dominates (Soboleva & Anthony, 2025; Jin et al., 2026). Future directions include communication-computation overlap and multi-node validation. This paper proposes and tests new approaches under an academic budget, and we hope to inspire production-scale adoption.

## Impact Statement

This paper presents work whose goal is to advance the field of Machine Learning. There are many potential societal consequences of our work, none which we feel must be specifically highlighted here.

## Acknowledgements

The authors acknowledge Research Computing at Arizona State University (Jennewein et al., 2023) for providing computing and storage resources that have contributed to the research results reported within this paper.

This work used Bridges-2 at Pittsburgh Supercomputing Center (Brown et al., 2021) through allocation CIS250914 from the Advanced Cyberinfrastructure Coordination Ecosystem: Services & Support (ACCESS) program, which is supported by National Science Foundation grants #2138259, #2138286, #2138307, #2137603, and #2138296.

This material is based upon work supported by the National Science Foundation Graduate Research Fellowship Program under Grant No. 2233001. Any opinions, findings, and conclusions or recommendations expressed in this material are those of the author(s) and do not necessarily reflect the views of the National Science Foundation.

We thank Keane te Velde for assistance with code development.

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

# A. Implementation Details

**Environments.** All experiments are conducted on NVIDIA H100 GPUs with 80GB, interconnected via NVLink.

**Baselines.** We compare Multi-Head LatentMoE with Head Parallel against MLP and MoE with Expert Parallel. MoE follows the definition in Section 2.2. For the MoE implementation, Grouped GEMM uses NVIDIA Transformer Engine (TE) (NVIDIA, 2022), token clustering uses TE's `moe_permute(.)` and `moe_sort_chunks_by_index(.)`, and Expert Parallel uses PyTorch's `all_to_all(.)`.

**Architecture.** All models are decoder-only Transformers with 12 layers, embedding size 1024, and context window 2048. We use rotary positional embeddings and RMSNorm without bias terms. The attention module has 8 heads with head dimension 128. Following (Liu et al., 2024), we replace the first two MoE layers with MLPs to mitigate router imbalance in early layers. All routers compute in FP32 for numerical stability, and we use aux-free (Wang et al., 2024a) and global (Qiu et al., 2025) load balancing.

**Initialization.** Following standard practices (Radford et al., 2019), we remove all biases, and all weights are randomly initialized from $\mathcal{N}(0, 0.02)$. Output projection weights are further scaled by $1/\sqrt{2 \cdot L}$ where L is the number of layers, also following Radford et al. (2019).

**Training.** We use AdamW with $\beta_1 = 0.9$, $\beta_2 = 0.95$, and weight decay 0.1. Learning rate is $5.0 \times 10^{-4}$ (following (Chi et al., 2022; Wang et al., 2024b; Wu et al., 2024)) with trapezoidal schedule of 8000 warmup steps and 2000 decay steps. The global batch size is 0.66 million tokens.

**Evaluation.** At training time, we estimate validation perplexity on a held-out set of 99.6 million tokens from the same distribution. After training, we use lm-eval-harness (Gao et al., 2024) to perform zero-shot evaluation on five downstream tasks: HellaSwag, PiQA, LAMBADA, ARC-Easy, and ARC-Challenge.

## B. Model Hyperparameters

We report hyperparameters for each experiment setting in Tables 5, 6, and 7. Unless otherwise specified, all models are decoder-only Transformers with RMSNorm (no bias), rotary positional embeddings (RoPE), and a context length of $T = 2048$.

**Training hyperparameters (shared).** All experiments use AdamW with $(\beta_1, \beta_2) = (0.9, 0.95)$, weight decay 0.1, learning rate $5.0 \times 10^{-4}$ with a trapezoidal schedule (8000 warmup steps, 2000 decay steps), and a global batch size of 0.66M tokens. All runs are conducted on NVIDIA H100 80GB GPUs interconnected via NVLink.

*Table 5.* Hyperparameters for the 0.2B–0.2B MLP baseline.

| Hyperparameter | MLP |
| --- | --- |
| Parameters (active–total) | 0.2B–0.2B |
| Transformer layers $L$ | 12 |
| Embedding size $d$ | 1024 |
| Attention heads | 8 |
| Head dimension | 128 |
| Context length $T$ | 2048 |
| Feedforward | Dense MLP |
| MoE experts $N_e$ | – |
| Active experts $k$ | – |
| Expert hidden size | – |
| Multi-head count $N_h$ | – |
| Latent head dim $d_h$ | – |

*Table 6.* Hyperparameters for the 0.2B–2.2B category.

| Hyperparameter | MoE (EP) | LatentMoE (EP) | MH LatentMoE (HP) | MH LatentMoE (HP, G) |
| --- | --- | --- | --- | --- |
| Parameters (active–total) | 0.2B–2.2B | 0.2B–2.2B | 0.2B–2.2B | 0.2B–2.2B |
| Transformer layers $L$ | 12 | 12 | 12 | 12 |
| Embedding size $d$ | 1024 | 1024 | 1024 | 1024 |
| Attention heads | 8 | 8 | 8 | 8 |
| Head dimension | 128 | 128 | 128 | 128 |
| Context length $T$ | 2048 | 2048 | 2048 | 2048 |
| MoE experts $N_e$ | 384 | 384 | 384 | 768 |
| Active experts $k$ | 4 | 4 | 4 | 8 |
| Expert hidden size | 256 | 256 | 256 | 128 |
| Multi-head count $N_h$ | – | – | 8 | 8 |
| Latent head dim $d_h$ | – | – | 128 | 128 |

*Table 7.* Hyperparameters for the 0.2B–4.2B category.

| Hyperparameter | MoE (EP) | LatentMoE (EP) | MH LatentMoE (HP) | MH LatentMoE (HP, G) |
|---|---|---|---|---|
| Parameters (active–total) | 0.2B–4.2B | 0.2B–4.2B | 0.2B–4.2B | 0.2B–4.2B |
| Transformer layers $L$ | 12 | 12 | 12 | 12 |
| Embedding size $d$ | 1024 | 1024 | 1024 | 1024 |
| Attention heads | 8 | 8 | 8 | 8 |
| Head dimension | 128 | 128 | 128 | 128 |
| Context length $T$ | 2048 | 2048 | 2048 | 2048 |
| MoE experts $N_e$ | 768 | 768 | 768 | 1536 |
| Active experts $k$ | 4 | 4 | 4 | 8 |
| Expert hidden size | 256 | 256 | 256 | 128 |
| Multi-head count $N_h$ | – | – | 8 | 8 |
| Latent head dim $d_h$ | – | – | 128 | 128 |

## C. 10B Token Experiments

Table 8 shows results at 10B tokens. At this scale, MH LatentMoE HP with double the granularity is the most competitive. MH LatentMoE HP trains 1.61× faster than MoE EP at the 4.2B scale (34.41 vs 55.34 hours), but does not yet surpass baselines in model quality. LatentMoE EP achieves the best perplexity at 2.2B parameters. However, when scaling to 50B tokens (Table 1 in main text), MH LatentMoE HP surpasses both MoE EP and LatentMoE EP across all configurations. This suggests our method benefits more from additional training data. This means ultra-sparse architectures are more data hungry.

*Table 8.* Model performance comparison across feedforward designs. Parameters are reported as activated-total. FineWebEDU reports validation perplexity (ppl.). HellaSwag (Hella.), PiQA, LAMBADA (LMB.), ARC-Easy (Arc E.), and ARC-Challenge (Arc C.) report zero-shot accuracy (acc.). G doubles the granularity. Bold indicates best and underline indicates second best. LatentMoE is replicated based on Elango et al. (2026). All accuracies are normalized based on length, except for LAMBADA.

| Parameters active - all | Model | Training Cost hours (rel.) ↓ | FineWebEDU ppl. ↓ | Hella. acc. ↑ | PiQA acc. ↑ | LMB. acc. ↑ | Arc E. acc. ↑ | Arc C. acc. ↑ | Avg. acc. ↑ |
|---|---|---|---|---|---|---|---|---|---|
| 0.2B-0.2B | MLP | 14.53 (1.00×) | 20.16 | 31.8 | 62.1 | 26.0 | 47.6 | 25.3 | 38.53 |
| 0.2B-2.2B | MoE EP | 35.36 (1.00×) | 15.56 | 39.7 | 67.1 | 32.4 | 53.1 | 27.6 | 43.95 |
| | LatentMoE EP | 32.71 (0.93×) | **15.31** | 40.6 | 67.7 | 33.1 | 53.5 | 29.3 | **44.83** |
| | **MH LatentMoE HP** | **31.68** (**0.90×**) | 15.61 | 39.5 | 67.0 | 31.2 | 53.7 | 28.2 | 43.93 |
| | **MH LatentMoE HP G** | 43.92 (1.24×) | 15.52 | 39.6 | 67.2 | 32.0 | 55.2 | 29.7 | 44.76 |
| 0.2B-4.2B | MoE EP | 55.34 (1.00×) | 15.01 | 41.2 | 67.6 | 33.3 | 54.6 | 29.8 | 45.30 |
| | LatentMoE EP | 53.76 (0.97×) | 14.84 | 41.8 | 68.1 | 32.7 | 52.5 | 27.6 | 44.55 |
| | **MH LatentMoE HP** | **34.41** (**0.62×**) | 15.02 | 41.2 | 67.8 | 33.1 | 54.3 | 29.4 | 45.17 |
| | **MH LatentMoE HP G** | 50.04 (0.90×) | **14.82** | 41.3 | 67.6 | 34.1 | 54.3 | 29.9 | **45.43** |

# D. Comparison with DeepEP

We compare training speed against DeepEP (Zhao et al., 2025b), a highly optimized Expert Parallel implementation with kernel fusion and communication-computation overlap. Table 9 shows the results estimated on 1000 training iterations.

DeepEP accelerates MoE training over our PyTorch EP baseline, but Multi-Head LatentMoE with HP is faster than both. Our EP and HP implementations are entirely in pure PyTorch, making the comparison fair and the code accessible.

*Table 9.* Training time comparison between PyTorch Expert Parallel, DeepEP, and our method. All experiments use 4 H100 GPUs. Relative speedup is computed against PyTorch EP. Model performance is identical across all three implementations up to 1000 iterations tested.

| Parameters | Method | Hours |
|---|---|---|
| 0.2B-2.2B | MoE PyTorch EP | 40.78 (1.00×) |
| | MoE DeepEP | 34.44 (1.18×) |
| | MH LatentMoE HP | 32.11 (1.27×) |
| 0.2B-4.2B | MoE PyTorch EP | 57.32 (1.00×) |
| | MoE DeepEP | 41.13 (1.39×) |
| | MH LatentMoE HP | 34.67 (1.65×) |

