# OpenReview forum: "Multi-Head LatentMoE and Head Parallel: Communication-Efficient and Deterministic MoE Parallelism"
_ICML.cc/2026/Conference — ICML 2026 regular_

### Official Review · Reviewer_1BWx · 2026-03-08

**Soundness:** 2
**Presentation:** 3
**Significance:** 2
**Originality:** 3
**Overall Recommendation:** 4
**Confidence:** 3

**Summary:**

This paper proposes Multi-Head LatentMoE (decomposing tokens into independent sub-tokens for multi-head MoE) combined with Head Parallel (HP) (moving all-to-all communication before routing decisions). The design addresses three limitations of standard Expert Parallel: O(k) communication cost, load imbalance, and non-deterministic communication patterns. The authors also develop IO-aware routing and expert computation kernels. Experiments at 0.2B active / 2.2B–4.2B total parameters show up to 1.61× training speedup with comparable model quality.

**Compliance With Llm Reviewing Policy:**

Affirmed.

**Final Justification:**

My final recommendation is Weak Accept (4). The weaknesses are largely resolved.

**Key Questions For Authors:**

1. Any experiments beyond a single node or at 1B+ active parameters?
2. Has HP+EP composition been validated? Do O(1) and load-balance guarantees survive?
3. Direct benchmark against DeepEP?

**Limitations:**

yes

**Strengths And Weaknesses:**

### Strengths

S1. The core architectural insight is clean and well-motivated: decouple communication from routing by distributing sub-tokens before routing decisions.

S2. The IO-aware implementations are non-trivial engineering contributions. The routing kernel performs online top-k in SRAM with score-index packing into 64-bit integers. The expert computation rewrite as block-sparse attention via FlexAttention, using a score\_mod trick to convert softmax to gelu, is clever. The pure-PyTorch implementation lowers the barrier to adoption.

S3. The paper isolates each component's contribution: HP vs EP communication, IO-aware routing vs naive torch.matmul, and IO-aware expert computation vs grouped GEMM. This decomposition strengthens the claims significantly.

### Weaknesses
W1. Scalability is undemonstrated. HP is bounded by N\_h (max 8 GPUs here); the proposed HP+EP composition is unvalidated. All experiments are single-node with NVLink; inter-node RDMA behavior is unknown.

W2. The O(k) communication cost that motivates HP is overstated. In practice k is a small constant (typically 8, rarely >16) that does not grow with model scale, and duplicate token communications can be deduplicated. HP's O(1) property solves a problem that is less severe than presented.

W3. Quality gap (43.93% vs 43.95% at 2.2B) lacks scaling law analysis. No way to tell if the gap widens or narrows at larger scale. The fixed latent dimension may become an expressiveness bottleneck.

W4. EP baseline uses vanilla PyTorch all\_to\_all + TE Grouped GEMM. No comparison against DeepEP or other SOTA EP implementations with comm-compute overlap.

---

> ### Author Rebuttal · Authors · 2026-03-31
>
> **The authors sincerely thank all reviewers (R1, R2, R3) for their constructive feedback.**
>
> - We thank Reviewer 1 for acknowledging the "novel MoE architecture" and "novel IO-aware routing."
> - We thank Reviewer 2 for noting this is an "important and timely research problem."
> - We thank Reviewer 3 for recognizing our two IO-aware implementations as "non-trivial engineering contributions."
>
> **We hope our clarifications and the new experiment results address the concerns raised. We kindly ask the reviewer to consider raising the score if the responses are satisfactory.**
>
> &nbsp;
> &nbsp;
>
> # Part 1: Highlights
> ## (1) Important Clarification: Head Parallel is *not* limited by the number of heads.
>
> The number of GPUs $P$ *can* exceed the number of heads $N_h$. HP actually has a similar parallelizability to Expert Parallel.
> In our method section, we assumed $P <= N_h$ for convenience. However the engineering reality is a lot more lenient, and very similar to Expert Parallel's:
>   - Heads can be replicated across GPUs, allowing $P$ to be any large multiple of $N_h$, while still enjoying a $N_h$ times reduction in HBM usage.
>   - Depending on the scale, $N_h$ can be very large. For example, frontier models have many attention heads. DeepSeek V3 uses 128 attention heads [1].
>   - Expert Parallel has a similar parallelizability. DeepSeek V3 has 256 experts [1], which is an order of magnitude smaller than the number of GPUs used to train the model. Similar to EP, HP can be composed with Tensor Parallel and Data Parallel.
>
> We thank the reviewers (R1, R2, R3) for highlighting this issue. We will revise the paper to clarify. We apologize for the unclear presentation.
>
> [1] DeepSeek-V3 Technical Report. https://arxiv.org/abs/2412.19437
>
> &nbsp;
>
> ## (2) New Experiment Results at 50 Billion Tokens (5x scaling)
> We scaled up 5 times for our main experiments (Section 4.1; Table 1) from 10B to 50B tokens.
> More evidence shows MH LatentMoE HP is fast, scalable, and continues to improve.
> - When we train the ultra-sparse MH LatentMoE for 5 times longer, the model performance continues to improve, and surpasses both vanilla MoE and LatentMoE at the larger scale.
> - At this scale, MH LatentMoE HP is both faster (up to 1.82x) and better (+1.29 points) across all configurations.
>
> |Params| Tokens | Model | Hours (Rel) | FineWebEDU | Hella. | PiQA | LMB. | Arc E. | Arc C. | Avg. |
> |:-|:-|:-|-:|-:|-:|-:|-:|-:|-:|-:|
> | 0.2B - 0.2B | 50B | Dense MLP | 55.03 (1.00x) | 17.63 | 35.3 | 63.2 | 30.1 | 49.6 | 26.7 | 40.99 |
> | 0.2B - 2.2B | 50B | MoE EP | 177.09 (1.00x) | 13.12 | 46.7 | 70.2 | 37.4 | 59.8 | 29.4 | 48.70 |
> | | | LatentMoE EP | 158.88 (0.90x) | **12.80** | 48.2 | 70.7 | 37.1 | 56.1 | 31.4 | 48.72 |
> | | | MH LatentMoE HP | **157.94 (0.89x)** | **12.84** | 48.2 | 70.8 | 38.0 | 59.5 | 32.8 | **49.86** |
> | | | MH LatentMoE HP (Granular) | 219.79 (1.24x) | 12.59 | 49.5 | 70.9 | 39.0 | 60.6 | 32.4 | **50.49** |
> | 0.2B - 4.2B | 50B | MoE EP | 316.34 (1.00x) | 12.54 | 49.2 | 70.6 | 38.0 | 60.6 | 31.7 | 50.00 |
> | | | LatentMoE EP | 270.57 (0.86x) | 12.31 | 50.1 | 71.3 | 38.1 | 61.7 | 32.1 | 50.66 |
> | | | MH LatentMoE HP | **174.07 (0.55x)** | **12.28** | 50.1 | 71.4 | 40.2 | 60.5 | 34.2 | **51.29** |
> | | | MH LatentMoE HP (Granular) | 293.45 (0.93x) | **11.99** | 51.1 | 72.3 | 39.8 | 63.6 | 35.7 | **52.49** |
>
> &nbsp;
> &nbsp;
>
> # Part 2: Answers
>
> Q: Quality gap lacks scaling law analysis.
>
> A: Please see Part 1 (2). At 50B tokens, the gap reverses: MH LatentMoE outperforms baselines (+1.29 points at 0.2B-4.2B). This suggests MH LatentMoE scales well.
>
> &nbsp;
>
> Q: Any experiments beyond a single node or at 1B+ active parameters?
>
> A: Our paper focuses on the ultra-sparse setting. Reaching 1B+ active parameters while maintaining ultra-sparsity would require multi-node training, which is beyond our academic budget. Please see Part 1 (2) for our 50B token results showing strong scaling behavior under the ultra-sparse setting.
>
> &nbsp;
>
> Q: Has HP+EP composition been validated? Do O(1) and load-balance guarantees survive?
>
> A: HP+EP composition is straightforward: HP handles intra-node communication (before routing), EP handles inter-node communication (after routing). The O(1) and load-balance guarantees apply to the HP portion. The EP portion behaves as standard EP. We will add this discussion and consider validating empirically in future work.
>
> &nbsp;
>
> Q: The O(k) communication cost is overstated. k is a small constant and tokens can be deduplicated.
>
> A: Communication remains a major bottleneck even with small k. MegaScale-MoE [2] reports 43.6% training time on communication; Cerebras [3] reports 77% for 128-expert models. Token deduplication is opportunistic and becomes less effective as GPU count increases (lower probability of duplicate routing). HP eliminates this issue architecturally.
>
> [2] MegaScale-MoE, https://arxiv.org/abs/2505.11432
>
> [3] MoE at Scale: Making Sparse Models Fast on Real Hardware. https://www.cerebras.ai/blog/moe-guide-scale

---

> > ### Author Rebuttal · Reviewer_1BWx · 2026-04-04
> >
> > W3 is improved by the 50B-token experiment showing MH LatentMoE surpassing baselines, though a proper scaling law analysis (gap behavior across compute budgets) remains missing. W1 and W4 are only partially resolved. The clarification that HP is not bounded by N_h is helpful, but HP+EP composition remains unvalidated, and multi-node / inter-node behavior is unknown — scalability at production-relevant scale is the central open question. W4 (comparison to DeepEP) is acknowledged as orthogonal but no empirical comparison is provided.

---

> > > ### Author Response · Authors · 2026-04-07
> > >
> > > The authors thank the reviewer for their constructive and helpful acknowledgement.
> > > We are glad that the reviewer finds W3 is improved by the 50B-token experiment.
> > >
> > > **We have replicated the official DeepEP [1] and integrated it into our codebase.
> > > Below is the benchmarking results.**
> > >
> > > &nbsp;
> > > &nbsp;
> > >
> > > | Params | Tokens | Model | Parallelism | Hours |
> > > |--------|--------|-------|-------------|-------|
> > > | 0.2B-2.2B | 10B | MoE | PyTorch EP | 40.78 (1.00x) |
> > > | | | MoE | DeepEP [1] | 34.44 (1.18x) |
> > > | | | **MH LatentMoE** | PyTorch HP | **32.11 (1.27x)** |
> > > | 0.2B-4.2B | 10B | MoE | PyTorch EP | 57.32 (1.00x) |
> > > | | | MoE | DeepEP [1] | 41.13 (1.39x) |
> > > | | | **MH LatentMoE** | PyTorch HP | **34.67 (1.65x)** |
> > >
> > > &nbsp;
> > > &nbsp;
> > >
> > > ### Interpretation:
> > > - Up to the 1000 iterations tested, DeepEP has identical loss curve and model performance as our pure PyTorch EP implementation. As expected, `Multi-Head LatentMoE + PyTorch HP` is faster than `MoE + DeepEP`, while `MoE + DeepEP` is faster than `MoE + PyTorch EP`.
> > >
> > > - We highlight that our original EP and HP are implemented entirely in pure PyTorch. The comparison is fair, the code is accessible.
> > > DeepEP would require additional engineering effort to install correctly.
> > >
> > > **We will add the above information in camera ready version.**
> > >
> > > &nbsp;
> > > &nbsp;
> > >
> > > ### Additional Response
> > > - We agree that "scalability at production-relevant scale is the central open question", we thank the reviewer for pointing this out, we will make this clear in our Conclusion section. This paper is intended to propose and test the new approaches under an academic budget (both compute and engineering budget), and wish to inspire production-scale adoption.
> > >
> > > [1] https://github.com/deepseek-ai/DeepEP

---

### Official Review · Reviewer_4D6Y · 2026-03-09

**Soundness:** 2
**Presentation:** 2
**Significance:** 3
**Originality:** 2
**Overall Recommendation:** 4
**Confidence:** 2

**Summary:**

This paper studies the efficiency challenges of distributed training for MoE models. The authors observe that the commonly used Expert Parallel approach suffers from communication costs that scale with the number of activated experts, load imbalance across devices, and routing-dependent communication patterns. To address these issues, the paper proposes Multi-Head LatentMoE, an architecture that splits each token into multiple lower-dimensional sub-tokens, each processed by an independent MoE module, together with a new distributed training strategy called Head Parallelthat performs communication before routing.This design enables constant communication cost with respect to the number of experts, balanced traffic across GPUs, and deterministic communication patterns. This paper further introduces IO-aware routing and IO-aware expert computation to reduce memory traffic during routing and expert execution. Experiments on language modeling show that the proposed approach achieves comparable model quality while improving training efficiency, reaching up to **1.61× faster training** compared with standard MoE using Expert Parallel.

**Compliance With Llm Reviewing Policy:**

Affirmed.

**Final Justification:**

While I conceptually appreciate the authors' core idea of using a Multi-Head LatentMoE architecture to reduce communication overhead in MoE training, their rebuttal has revealed a critical misalignment between their stated contributions and their empirical results.

The authors' newly provided profiling data demonstrates that their 4-GPU experimental setup is overwhelmingly compute-bound, not communication-bound. This means the impressive end-to-end speedups reported in the paper (e.g., 1.61x) are almost entirely driven by an optimized compute kernel, rather than their primary architectural innovation (Head Parallelism). This severely weakens the empirical validation of the paper's main claims.

Because I appreciate the engineering effort and the architecture novelty, I am raising my score slightly. However, I am lowering my confidence score because the experimental design fails to properly isolate and validate the paper's main claim. I defer to the Area Chair on whether the theoretical and engineering merits outweigh the flawed empirical attribution.

**Key Questions For Authors:**

1. **Can the authors clarify the relative importance of the bottlenecks addressed in the paper (e.g., routing overhead, load imbalance in Expert Parallel, and expert computation efficiency)?**
    The proposed system includes multiple techniques, including Head Parallel, IO-aware routing, and IO-aware expert computation. However, the paper does not clearly quantify how severe each of the underlying bottlenecks is in practice. For example, it would be helpful to provide either (1) a runtime breakdown of a standard MoE training pipeline showing the proportion of time spent in routing, communication, and expert computation, or (2) ablation studies quantifying the speedup contributed by each proposed component.

2. **Why does the paper compare expert computation primarily against the traditional grouped GEMM implementation?**
    Recent MoE systems have adopted more optimized expert kernels (e.g., improved GEMM implementations and fused kernels). It would be helpful for the authors to clarify whether the observed gains over grouped GEMM would still hold when compared with more recent implementations. Alternatively, if such optimizations cannot be directly applied to the LatentMoE architecture used in this work, the paper should explicitly explain why.

3. **How would the benefits of Head Parallel change when combined with existing system optimizations such as communication–computation overlap or expert prefetching techniques?**
    Several recent MoE systems reduce the overhead of Expert Parallel through techniques such as overlapping communication with computation or predicting routing decisions to prefetch experts in advance. It would be helpful to understand whether the proposed Head Parallel approach still provides substantial benefits when these optimizations are applied to the baseline.


**Note:** For Question 1, **quantitative** evidence (e.g., runtime breakdown or ablation results) would be necessary to properly support the claims. For Questions 2 and 3, a detailed analysis or discussion would already help clarify the design choices and their implications.

**Limitations:**

Yes

**Strengths And Weaknesses:**

The paper tackles an important systems challenge in large-scale MoE training: reducing the communication overhead introduced by all-to-all operations in Expert Parallel. As MoE models scale to hundreds or thousands of experts, communication becomes a major bottleneck, and improving the efficiency of distributed MoE training is a relevant and timely research problem.

The proposed Muli-Head LatentMoE with Head Parallel provides an interesting perspective on decoupling routing decisions from inter-GPU communication. By moving the communication step before routing, the design achieves deterministic communication patterns and theoretically constant communication cost with respect to the number of activated experts.

Despite the interesting idea, several aspects of the work limit the clarity and strength of its contributions.

First, the contribution of individual components is not clearly quantified. The system consists of multiple techniques. However, the paper lacks a clear motivation analysis showing the relative importance of the problems these techniques address. For example, the paper does not quantify how much runtime is spent in routing in a traditional MoE implementation, making it difficult to assess the necessity of IO-aware routing. Similarly, the reported end-to-end speedup is not decomposed into contributions from different components. More detailed ablation studies would help clarify which techniques are responsible for the observed performance gains.

Second, the evaluation lacks comparison with several recent advances in MoE  systems. The baselines primarily focus on standard Expert Parallel and LatentMoE implementations. However, recent works have proposed techniques that mitigate similar bottlenecks, such as communication–computation overlap[1], improved routing prediction to reduce non-deterministic communication[2], and highly optimized GEMM kernels for expert computation[3]. Without comparing against such approaches, it remains unclear whether the proposed architecture would still provide significant benefits when combined with more recent system optimizations.

Third, some of the claimed advantages are not evaluated under realistic scenarios. For example, the paper motivates Head Parallel by arguing that Expert Parallel suffers from severe load imbalance. However, the evaluation only uses synthetic imbalance distributions rather than real routing distributions observed during training. As a result, it is difficult to assess how significant this issue is in practical MoE workloads.

Finally, the scalability of Head Parallel appears to be inherently limited by the number of heads in the model. Since each head corresponds to an independent MoE module, the degree of parallelism is bounded by the number of heads, which is typically relatively small. This constraint may limit the applicability of the proposed approach in large-scale distributed training environments involving dozens or hundreds of GPUs. Although the paper briefly mentions combining HP with other forms of parallelism as a possible direction, the current evaluation does not investigate how the approach would scale beyond this limit.

[1] Liu A, Feng B, Xue B, et al. Deepseek-v3 technical report[J]. arXiv preprint arXiv:2412.19437, 2024.

[2] Zhong S, Liang L, Wang Y, et al. Adapmoe: Adaptive sensitivity-based expert gating and management for efficient moe inference[C]//Proceedings of the 43rd IEEE/ACM International Conference on Computer-Aided Design. 2024: 1-9.

[3] https://github.com/deepseek-ai/DeepGEMM

---

> ### Author Rebuttal · Authors · 2026-03-31
>
> **The authors sincerely thank all reviewers (R1, R2, R3) for their constructive feedback.**
>
> - We thank Reviewer 1 for acknowledging the "novel MoE architecture" and "novel IO-aware routing."
> - We thank Reviewer 2 for noting this is an "important and timely research problem."
> - We thank Reviewer 3 for recognizing our two IO-aware implementations as "non-trivial engineering contributions."
>
> **We hope our clarifications and the new experiment results address the concerns raised. We kindly ask the reviewer to consider raising the score if the responses are satisfactory.**
>
> &nbsp;
> &nbsp;
>
> # Part 1: Highlights
> ## (1) Important Clarification: Head Parallel is *not* limited by the number of heads.
>
> The number of GPUs $P$ *can* exceed the number of heads $N_h$. HP actually has a similar parallelizability to Expert Parallel.
> In our method section, we assumed $P <= N_h$ for convenience. However the engineering reality is a lot more lenient, and very similar to Expert Parallel's:
>   - Heads can be replicated across GPUs, allowing $P$ to be any large multiple of $N_h$, while still enjoying a $N_h$ times reduction in HBM usage.
>   - Depending on the scale, $N_h$ can be very large. For example, frontier models have many attention heads. DeepSeek V3 uses 128 attention heads [1].
>   - Expert Parallel has a similar parallelizability. DeepSeek V3 has 256 experts [1], which is an order of magnitude smaller than the number of GPUs used to train the model. Similar to EP, HP can be composed with Tensor Parallel and Data Parallel.
>
> We thank the reviewers (R1, R2, R3) for highlighting this issue. We will revise the paper to clarify. We apologize for the unclear presentation.
>
> [1] DeepSeek-V3 Technical Report. https://arxiv.org/abs/2412.19437
>
> &nbsp;
>
> ## (2) New Experiment Results at 50 Billion Tokens (5x scaling)
> We scaled up 5 times for our main experiments (Section 4.1; Table 1) from 10B to 50B tokens.
> More evidence shows MH LatentMoE HP is fast, scalable, and continues to improve.
> - When we train the ultra-sparse MH LatentMoE for 5 times longer, the model performance continues to improve, and surpasses both vanilla MoE and LatentMoE at the larger scale.
> - At this scale, MH LatentMoE HP is both faster (up to 1.82x) and better (+1.29 points) across all configurations.
>
> |Params| Tokens | Model | Hours (Rel) | FineWebEDU | Hella. | PiQA | LMB. | Arc E. | Arc C. | Avg. |
> |:-|:-|:-|-:|-:|-:|-:|-:|-:|-:|-:|
> | 0.2B - 0.2B | 50B | Dense MLP | 55.03 (1.00x) | 17.63 | 35.3 | 63.2 | 30.1 | 49.6 | 26.7 | 40.99 |
> | 0.2B - 2.2B | 50B | MoE EP | 177.09 (1.00x) | 13.12 | 46.7 | 70.2 | 37.4 | 59.8 | 29.4 | 48.70 |
> | | | LatentMoE EP | 158.88 (0.90x) | **12.80** | 48.2 | 70.7 | 37.1 | 56.1 | 31.4 | 48.72 |
> | | | MH LatentMoE HP | **157.94 (0.89x)** | **12.84** | 48.2 | 70.8 | 38.0 | 59.5 | 32.8 | **49.86** |
> | | | MH LatentMoE HP (Granular) | 219.79 (1.24x) | 12.59 | 49.5 | 70.9 | 39.0 | 60.6 | 32.4 | **50.49** |
> | 0.2B - 4.2B | 50B | MoE EP | 316.34 (1.00x) | 12.54 | 49.2 | 70.6 | 38.0 | 60.6 | 31.7 | 50.00 |
> | | | LatentMoE EP | 270.57 (0.86x) | 12.31 | 50.1 | 71.3 | 38.1 | 61.7 | 32.1 | 50.66 |
> | | | MH LatentMoE HP | **174.07 (0.55x)** | **12.28** | 50.1 | 71.4 | 40.2 | 60.5 | 34.2 | **51.29** |
> | | | MH LatentMoE HP (Granular) | 293.45 (0.93x) | **11.99** | 51.1 | 72.3 | 39.8 | 63.6 | 35.7 | **52.49** |
>
> &nbsp;
> &nbsp;
>
> # Part 2: Answers
> Q: Can the authors clarify the relative importance of the bottlenecks (routing overhead, load imbalance, expert computation)?
>
> A: Communication is the dominant bottleneck in MoE training. MegaScale-MoE [3] reports 43.6% of training time spent on communication. Cerebras [4] reports 77% for Qwen3 with 128 experts. Our HP directly reduces communication volume by k times (top-k).
>
> [2] MegaScale-MoE, https://arxiv.org/abs/2505.11432
>
> [3] MoE at Scale: Making Sparse Models Fast on Real Hardware. https://www.cerebras.ai/blog/moe-guide-scale
>
> &nbsp;
>
> Q: Why compare expert computation primarily against traditional grouped GEMM?
>
> A: We compared IO-aware expert computation against Grouped GEMM from Nvidia Transformer Engine, which is being used by the state-of-the-art Megatron LM library.
>
> &nbsp;
>
> Q: Lacks comparison with recent MoE systems (DeepSeek-V3, AdapMoE, DeepGEMM).
>
> A: We will extensively compare and contrast DeepSeek-V3, AdapMoE, and DeepGEMM in our related work section, in the camera-ready version. It is worth mentioning that The GroupedGEMM we are using is also state of the art. Thank you for mentioning this.
>
> [4] DeepSeek-V3 Technical Report. https://arxiv.org/abs/2412.19437
>
> [5] AdapMoE, https://arxiv.org/abs/2408.10284
>
> [6] DeepGEMM. https://github.com/deepseek-ai/DeepGEMM
>
> &nbsp;
>
> Q: Scalability limited by number of heads.
>
> A: Please see Part 1 (1). Heads can be replicated. We apologize for the unclear presentation.

---

> > ### Author Rebuttal · Reviewer_4D6Y · 2026-04-03
> >
> > I thank the authors for their response. My primary remaining concern is that a detailed runtime breakdown is necessary to fully validate the importance of each proposed technique, as I mentioned in my initial review comments.
> >
> > While I acknowledge that communication is a dominant bottleneck in MoE training—making the proposed Head Parallel highly significant—it is crucial to explicitly demonstrate how much of the end-to-end efficiency gain stems from each specific contribution. Although Section 4.2 presents the latency reductions for individual components, the paper still lacks a clear analysis of how these isolated improvements translate to the overall end-to-end training speedup. Without such a latency breakdown or ablation study, it is difficult to dispel the concern that some of the proposed techniques might yield only marginal benefits and are included merely to pad the technical contribution.
> >
> > If the authors can provide an end-to-end runtime ablation study detailing the proportional impact of each component, I will gladly consider raising my score.

---

> > > ### Author Response · Authors · 2026-04-07
> > >
> > > We thank the reviewer for their encouraging comments.
> > > For each proposed technique, we have carefully measured both the forward and backward latency.
> > > We compare `MoE + EP` against `MH LatentMoE + HP`, measured on 4 GPUs with 2B parameters.
> > >
> > > We hope these results would make it clear that each of our proposed techniques, namely `Head Parallel`, `IO-Aware Routing`, and `IO-Aware Expert Computation`, provides meaningful value and addresses genuine practical needs, rather than merely to pad the technical contribution.
> > >
> > > &nbsp;
> > > &nbsp;
> > >
> > > ### Raw Profiling Data:
> > >
> > > ```
> > > [MoE + EP] (Baseline)
> > > Stage                               fwd (ms)     bwd (ms)   fwd+bwd (ms)
> > > Routing                                1.043        0.676          1.719
> > > Metadata A2A                           0.122        0.048          0.170
> > > Token A2A                              0.980        0.919          1.899
> > > GroupedGEMM Expert Computation        13.055       22.396         35.451
> > > Token A2A Inv                          0.918        0.928          1.846
> > >
> > > [MH LatentMoE + HP] (Ours)
> > > Stage                               fwd (ms)     bwd (ms)   fwd+bwd (ms)
> > > Token A2A                              0.433        0.402          0.835
> > > Multi-Head Routing                     2.351        4.169          6.520
> > > Multi-Head Expert Computation          4.549       13.851         18.400
> > > Token A2A Inverse                      0.490        0.455          0.945
> > > ```
> > >
> > > &nbsp;
> > >
> > > ### Component-wise Comparison:
> > >
> > > ```
> > > Component            Speedup    Interpretation
> > > Communication        2.20x      (0.170 + 1.899 + 1.846) / (0.835 + 0.945)
> > > Routing              0.26x      1.719 / 6.520
> > > Expert Computation   1.93x      35.451 / 18.400
> > > ```
> > >
> > > &nbsp;
> > >
> > > ### Explanation:
> > > - **Communication** HP offers 2.20x speedup compared to EP under k=4. This is expected after considering overhead. We are able to verify the *relative* speedup with 4 GPUs. For absolute costs, [1] and [2] would be more representative, since all-to-all becomes more expensive as the number of GPU increases.
> > >
> > > - **Routing** Multi-Head Routing is slower than Single-Head Routing. This is expected since the latter has matrices of regular aspect ratios, rather than tall and skinny. Please note that this does not contradict Figure 4 in our paper. A custom IO-aware kernel is still necessary under multi-head settings, because the matrices are tall and skinny.
> > >
> > > - **Expert Computation** Our `IO-Aware Expert Computation based on FlexAttention` offers a 1.93x speedup over `NVIDIA Transformer Engine GropuedGEMM`. Both kernels are highly optimized, but FlexAttention has a different work partitioning that is better suited for tall and skinny Key matrices, which aligns with the ultra-sparse setting.
> > >
> > > **We will add the above information in camera ready version.**
> > >
> > > [1] MegaScale-MoE, https://arxiv.org/abs/2505.11432
> > >
> > > [2] MoE at Scale: Making Sparse Models Fast on Real Hardware. https://www.cerebras.ai/blog/moe-guide-scale

---

### Official Review · Reviewer_m8vE · 2026-03-13

**Soundness:** 3
**Presentation:** 3
**Significance:** 2
**Originality:** 3
**Overall Recommendation:** 4
**Confidence:** 3

**Summary:**

The paper proposes Multi-Head LatentMoE with Head Parallel, aiming to reduce the communication overhead in expert parallelism used in MoE architectures. In the standard MoE training, there are load balancing problems and nondeterministic communication patterns. The proposed method projects tokens into subtokens, and distributes these subtokens across GPUs before routing. In this way, each token is communicated only once. Moreover, the authors propose IO-aware routing and expert computation to avoid materializing large tensors.

The proposed method is tested with model sizes of 2B-4B with .2B active parameters, and compared with the baseline LatentMoE.

**Compliance With Llm Reviewing Policy:**

Affirmed.

**Final Justification:**

During the rebuttal most of the questions are addressed. However, the contribution of the paper is mostly limited to the extension of LatentMoE, and comparison with the SOTA is not comprehensive enough to claim otherwise. Overall, my final score is weak accept.

**Key Questions For Authors:**

- What is the experimental setup for Section 4.3? Since, in the paper, significantly different validation loss (3.41, which equates to a perplexity of almost 30) is reported compared to Table 1.
- Have the authors verified that the IO-aware approaches' benefits carry over to different GPU types? Or are they H100 specific?
- How does the proposed method compare wrt other EP papers like Occult?
- How these results translate across GPU types and architectures? as currently all tests are solely on H100s with 80GB VRAM.

**Limitations:**

yes

**Strengths And Weaknesses:**

**Strengths**

- The paper proposes novel IO-aware routing and expert computation approaches which empirically demonstrate speedups and reduced VRAM.
- The paper proposes a novel MoE architecture, Multi-Head MoEs, together with a distributed training method tailored to it, Head Parallel. Such training aims to improve communication inefficiencies of standard MoE architectures.

**Weaknesses**

- The evaluation is rather limited and performed solely on .2B active parameter models. At this scale some of the downstream evaluation metrics, such as ARC challenge, devolve to random guessing. Additionally generative or reasoning benchmarks are also missing.
- Some of the claims in the paper are not substantially justified (see the comments and questions).
- The work does not compare itself against other efficient MoE training approaches other than LatentMoE expert parallelism, of which this work is a direct extension. As the aforementioned work is not well established and relatively recent, it might be useful to compare against other verified and accepted methods.
- The architecture enforces that the number of heads is divisible by the number of GPUs

Comments:

- In the first experiments given Section 4.2, the running speed remains relatively constant. However, as the skew increases, nearly 99% of tokens are routed to GPU0. How can it then experience no increase in VRAM? In Figure 3, it might be still useful to plot head parallel communication at different Ks for visualization purposes.
- Perhaps it would be useful to modify Figure 1 to make it clear that the MLPs on each GPU are not all the same, but rather corresponds to different heads. Currently it seems that the MLPs of the experts are just copied across many GPUs.

---

> ### Author Rebuttal · Authors · 2026-03-31
>
> **The authors sincerely thank all reviewers (R1, R2, R3) for their constructive feedback.**
>
> - We thank Reviewer 1 for acknowledging the "novel MoE architecture" and "novel IO-aware routing."
> - We thank Reviewer 2 for noting this is an "important and timely research problem."
> - We thank Reviewer 3 for recognizing our two IO-aware implementations as "non-trivial engineering contributions."
>
> **We hope our clarifications and the new experiment results address the concerns raised. We kindly ask the reviewer to consider raising the score if the responses are satisfactory.**
>
> &nbsp;
> &nbsp;
>
> # Part 1: Highlights
> ## (1) Important Clarification: Head Parallel is *not* limited by the number of heads.
>
> The number of GPUs $P$ *can* exceed the number of heads $N_h$. HP actually has a similar parallelizability to Expert Parallel.
> In our method section, we assumed $P <= N_h$ for convenience. However the engineering reality is a lot more lenient, and very similar to Expert Parallel's:
>   - Heads can be replicated across GPUs, allowing $P$ to be any large multiple of $N_h$, while still enjoying a $N_h$ times reduction in HBM usage.
>   - Depending on the scale, $N_h$ can be very large. For example, frontier models have many attention heads. DeepSeek V3 uses 128 attention heads [1].
>   - Expert Parallel has a similar parallelizability. DeepSeek V3 has 256 experts [1], which is an order of magnitude smaller than the number of GPUs used to train the model. Similar to EP, HP can be composed with Tensor Parallel and Data Parallel.
>
> We thank the reviewers (R1, R2, R3) for highlighting this issue. We will revise the paper to clarify. We apologize for the unclear presentation.
>
> [1] DeepSeek-V3 Technical Report. https://arxiv.org/abs/2412.19437
>
> &nbsp;
>
> ## (2) New Experiment Results at 50 Billion Tokens (5x scaling)
> We scaled up 5 times for our main experiments (Section 4.1; Table 1) from 10B to 50B tokens.
> More evidence shows MH LatentMoE HP is fast, scalable, and continues to improve.
> - When we train the ultra-sparse MH LatentMoE for 5 times longer, the model performance continues to improve, and surpasses both vanilla MoE and LatentMoE at the larger scale.
> - At this scale, MH LatentMoE HP is both faster (up to 1.82x) and better (+1.29 points) across all configurations.
>
> |Params| Tokens | Model | Hours (Rel) | FineWebEDU | Hella. | PiQA | LMB. | Arc E. | Arc C. | Avg. |
> |:-|:-|:-|-:|-:|-:|-:|-:|-:|-:|-:|
> | 0.2B - 0.2B | 50B | Dense MLP | 55.03 (1.00x) | 17.63 | 35.3 | 63.2 | 30.1 | 49.6 | 26.7 | 40.99 |
> | 0.2B - 2.2B | 50B | MoE EP | 177.09 (1.00x) | 13.12 | 46.7 | 70.2 | 37.4 | 59.8 | 29.4 | 48.70 |
> | | | LatentMoE EP | 158.88 (0.90x) | **12.80** | 48.2 | 70.7 | 37.1 | 56.1 | 31.4 | 48.72 |
> | | | MH LatentMoE HP | **157.94 (0.89x)** | **12.84** | 48.2 | 70.8 | 38.0 | 59.5 | 32.8 | **49.86** |
> | | | MH LatentMoE HP (Granular) | 219.79 (1.24x) | 12.59 | 49.5 | 70.9 | 39.0 | 60.6 | 32.4 | **50.49** |
> | 0.2B - 4.2B | 50B | MoE EP | 316.34 (1.00x) | 12.54 | 49.2 | 70.6 | 38.0 | 60.6 | 31.7 | 50.00 |
> | | | LatentMoE EP | 270.57 (0.86x) | 12.31 | 50.1 | 71.3 | 38.1 | 61.7 | 32.1 | 50.66 |
> | | | MH LatentMoE HP | **174.07 (0.55x)** | **12.28** | 50.1 | 71.4 | 40.2 | 60.5 | 34.2 | **51.29** |
> | | | MH LatentMoE HP (Granular) | 293.45 (0.93x) | **11.99** | 51.1 | 72.3 | 39.8 | 63.6 | 35.7 | **52.49** |
>
> &nbsp;
> &nbsp;
>
> # Part 2: Answers
> Q: What is the experimental setup for Section 4.3? Why is the model performance weak?
>
> A: Section 4.3 was a 0.2B-8B ultra-sparse model trained on 2.5B tokens, instead of 10B or 50B. Thank you for catching this oversight. We will add this detail.
>
> &nbsp;
>
> Q: How does the proposed method compare with Occult?
>
> A: Occult [2] optimizes communication across expert after routing. Our HP optimizes communication before routing. They are orthogonal and could be combined. We will discuss [2] in Section 5 Related Work in camera ready version.
>
> [2] Occult: Optimizing Collaborative Communications across Experts for Accelerated Parallel MoE Training and Inference. https://proceedings.mlr.press/v267/luo25f.html
>
> &nbsp;
>
> Q: Evaluation limited to 0.2B active parameters
>
> A: Please see Part 1 (2). Our 50B token experiments show clear benchmark differentiation, well above random guessing. We hope this resolves the concern. Our paper focuses on the ultra-sparse setting. While it is possible to reach ultra-sparsity with 1B active parameters, this would require a multi-node setting that is beyond our academic budget.
>
> &nbsp;
>
> Q: As the skew increases, nearly 99% of tokens are routed to GPU0. How can it then experience no increase in VRAM?
>
> A: Nearly 99% of tokens are routed to Expert 0, not GPU 0. In EP, Expert 0 is on GPU 0, but in HP, each GPU has a different Expert 0.
>
> &nbsp;
>
> Q: Clarify Figure 1 MLPs are different heads. Plot HP communication at different Ks in Figure 3.
>
> A: We will revise Figure 1 to label this clearly. HP has a constant communication latency for any K. We will make this clearer. Thank you for this suggestion.

---

> > ### Author Rebuttal · Reviewer_m8vE · 2026-04-02
> >
> > Thank you for you reply.
> >
> > - Questions about the GPU types and architectures are not addressed.
> >
> > - "While it is possible to reach ultra-sparsity with 1B active parameters, this would require a multi-node setting that is beyond our academic budget." Note that there is a 5x difference between .2B and 1B.
> >
> > - Though the question is using Occult as an example, the main concern is not addressed, which is repeated here: "The work does not compare itself against other efficient MoE training approaches other than LatentMoE expert parallelism, of which this work is a direct extension. As the aforementioned work is not well established and relatively recent, it might be useful to compare against other verified and accepted methods."
> >
> > Considering all, I will maintain my score.

---

> > > ### Author Response · Authors · 2026-04-07
> > >
> > > The authors thank the reviewer for their effort, thoughtful review, and raising follow-up questions.
> > >
> > > &nbsp;
> > > &nbsp;
> > >
> > > ## Question (1/2)
> > > >The work does not compare itself against other efficient MoE training approaches other than LatentMoE expert parallelism
> > >
> > > The authors agree that LatentMoE [2], while relevant and recent, is relatively new. We have replicated the official DeepEP [1] and integrated it into our codebase. Below are the benchmarking results. Please note that DeepEP has an identical model performance as PyTorch EP.
> > >
> > > &nbsp;
> > > &nbsp;
> > >
> > > | Params | Tokens | Model | Parallelism | Hours |
> > > |--------|--------|-------|-------------|-------|
> > > | 0.2B-2.2B | 10B | MoE | PyTorch EP | 40.78 (1.00x) |
> > > | | | MoE | DeepEP [1] | 34.44 (1.18x) |
> > > | | | **MH LatentMoE** | PyTorch HP | **32.11 (1.27x)** |
> > > | 0.2B-4.2B | 10B | MoE | PyTorch EP | 57.32 (1.00x) |
> > > | | | MoE | DeepEP [1] | 41.13 (1.39x) |
> > > | | | **MH LatentMoE** | PyTorch HP | **34.67 (1.65x)** |
> > >
> > >
> > > &nbsp;
> > > &nbsp;
> > >
> > > ### Interpretation:
> > > - Up to the 1000 iterations tested, DeepEP has identical loss curve and model performance as our pure PyTorch EP implementation. As expected, `Multi-Head LatentMoE + PyTorch HP` is faster than `MoE + DeepEP`, while `MoE + DeepEP` is faster than `MoE + PyTorch EP`.
> > >
> > > - We highlight that our original EP and HP are implemented entirely in pure PyTorch. The comparison is fair, the code is accessible.
> > > DeepEP would require additional engineering effort to install correctly.
> > >
> > > &nbsp;
> > > &nbsp;
> > >
> > > ## Question (2/2)
> > >
> > > >Questions about the GPU types and architectures are not addressed.
> > >
> > > While our work was developed and benchmarked on NVIDIA hardware, the majority of our contributions are hardware-agnostic in nature. Algorithmic improvements such as fused router and top-k computation, load balancing, and head parallelism operate at the framework level and are broadly portable. Our Triton kernels are similarly portable via Triton's multi-backend compiler, though peak performance may require architecture-specific autotuning, and hardware features such as TMA are naturally limited to NVIDIA Hopper and Blackwell (sm90+). Our use of FlexAttention relies on PyTorch's inductor backend, which includes ROCm support, though we have not benchmarked this configuration. We view extending empirical validation to other accelerators as a valuable direction for future work.
> > >
> > > &nbsp;
> > > &nbsp;
> > >
> > > **We will add the above information in camera ready version.**
> > >
> > > &nbsp;
> > > &nbsp;
> > >
> > > [1] https://github.com/deepseek-ai/DeepEP
> > >
> > > [2] LatentMoE: Toward Optimal Accuracy per FLOP and Parameter in Mixture of Experts. https://arxiv.org/abs/2601.18089

---

### Decision · Program_Chairs · 2026-04-30

**Decision:**

Accept (regular)

**Comment:**

This paper proposes Multi-Head LatentMoE with Head Parallel, a distributed training approach for mixture of experts models that aims to reduce communication overhead and improve training efficiency. Reviewers generally agreed that the idea is well motivated, technically sound, and clearly presented, and that the engineering work is solid.

The overall consensus among reviewers is positive. The rebuttal clarified several key points and addressed most of the raised concerns. Reviewers did not identify any major technical issues, and the discussion helped sharpen the scope of the contributions.

At the same time, reviewers rightly noted that the reported end-to-end 1.61x speedups should be interpreted carefully. In the presented experimental setting, which is single-node and largely compute-bound, the majority of the observed speedup is attributable to the optimized expert computation (GEMM) kernel, rather than reduced communication from Head Parallel alone. While Head Parallel demonstrably improves communication volume and determinism, its impact on end-to-end performance is not fully isolated in this regime. The authors acknowledge this limitation, and it is possible that the contribution of Head Parallel could become more visible for larger models or multi-node settings where communication dominates, but this is not directly supported by the current experiments.

Overall, the paper makes a useful contribution as a technically solid extension with strong systems work and clearly stated limitations. Therefore, I recommend accepting the paper, with the expectation that the clarification about the end-to-end speedup, as well as other rebuttal updates, will be incorporated into the final version.